# Stakeholder-Identified Interventions to Address Cancer Survivors' Psychosocial Needs after Completing Treatment

**Sarah Murnaghan** [1], **Cynthia Kendell** [2], **Jonathan Sussman** [3], **Geoffrey A. Porter** [1,2], **Doris Howell** [4], **Eva Grunfeld** [5,6] **and Robin Urquhart** [1,2,*]

1. Department of Community Health and Epidemiology, Dalhousie University, Halifax, NS B3H 1V7, Canada; sr438725@dal.ca (S.M.); geoff.porter@dal.ca (G.A.P.)
2. Department of Surgery, Nova Scotia Health, Halifax, NS B3H 2Y9, Canada; cynthia.kendell@ccns.nshealth.ca
3. Department of Oncology, McMaster University, Hamilton, ON L8V 5C2, Canada; sussman@HHSC.CA
4. Department of Supportive Care, Princess Margaret Cancer Research Institute, Toronto ON M5G 0A3, Canada; doris.howell@uhn.ca
5. Ontario Institute for Cancer Research, Toronto ON M5G 0A3, Canada; eva.grunfeld@utoronto.ca
6. Department of Family and Community Medicine, University of Toronto, Toronto ON M5G 1V7, Canada
* Correspondence: robin.urquhart@nshealth.ca; Tel.: +902-473-7290; Fax: +902-473-4631

**Abstract:** The interventions used in cancer-survivorship care do not always address outcomes important to survivors. This study sought to understand stakeholders' views on the key concerns of cancer survivors after treatment and the interventions needed to meet survivors' and families' psychosocial needs after completing cancer treatment. We conducted a descriptive qualitative study using semi-structured interviews with stakeholders (survivors, family/friend caregivers, oncology providers, primary care providers, and cancer system decision-/policy-makers) from across Canada. For the data analysis, we used techniques commonly employed in descriptive qualitative research, such as coding, grouping, detailing, and comparing the data. There were 44 study participants: 11 survivors, seven family/friend caregivers, 18 health care providers, and eight decision-/policy-makers. Stakeholder-relevant interventions to address survivors' psychosocial needs were categorized into five groups, as follows: information provision, peer support, navigation, knowledge translation interventions, and caregiver-specific supports. These findings, particularly interventions that deliver timely and relevant information about the post-treatment period and knowledge translation interventions that strive to integrate effective tools and programs into survivorship care, have implications for future research and practice.

**Keywords:** cancer survivorship; psychosocial needs; interventions; qualitative methods

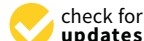



## 1. Background

The number of cancer survivors has risen dramatically over the past two decades, and 63% of all people diagnosed in Canada today will live for five years or more after their diagnosis [1]. This represents tremendous success in early detection and new therapies. At the same time, many cancer survivors experience substantial late and long-term adverse impacts due to cancer and its treatment. A large Canada-wide study found that 9 in 10 cancer survivors, in the 1–3 years after completing treatment, report ongoing physical needs, and nearly 8 in 10 report ongoing psychosocial needs [2]. The most prevalent of these needs were fatigue, changes in sexual function, and changes in memory and concentration (physical); worry about cancer recurrence, changes in sexual intimacy, and depression (emotional); and challenges presented when returning to work/school, challenges getting to/from appointments, and paying for health care (practical). These ongoing issues can negatively impact survivors' functional ability, survival, quality of life and their economic, emotional, and social wellbeing [3–9].

Much research in Canada and globally has demonstrated that cancer survivors experience distress after completing treatment. Indeed, survivors often describe the transition

from active treatment to routine follow-up care as a time filled with fear, distress, and feelings of abandonment [10–12]. Survivors feel unprepared in terms of the ongoing psychological and emotional burden they carry after treatment [10,13,14]. The Pan-Canadian Framework for Cancer Survivorship Research has prioritized the need for intervention research that specifically focuses on survivors' psychosocial needs, including interventions to support their transition back to daily life [15].

In this study, we sought to understand stakeholders' views on the key concerns of cancer survivors after treatment and the interventions that are needed to address these concerns. The aim was to inform future intervention research in cancer survivorship by identifying both the important outcomes and potential interventions. This paper presents stakeholders' views on the interventions needed to meet survivors' and families' psychosocial needs after completing cancer treatment.

## 2. Methods

We conducted a descriptive qualitative study [16] using semi-structured interviews with stakeholders (survivors, family/friend caregivers, oncology providers, primary care providers, and cancer system decision-/policy-makers) from across Canada. Purposive sampling was used to achieve variations in the sample with regard to cancer type, place of work or residence (urban/rural/remote), and special populations (e.g., pediatric, adolescent, and young adults).

### 2.1. Recruitment

Health care providers (oncology providers, primary care providers) and cancer system decision-/policy-makers were recruited via cancer programs/clinics; provincial and national organizations, groups, and networks (e.g., Canadian Partnership Against Cancer, provincial cancer agencies); and publicly available information from each jurisdiction (e.g., leads of primary care and oncology networks). To achieve a variation in the sampling characteristics, the research team identified and contacted individuals who specialized in different cancer types, who worked in urban and rural settings, and who worked with or had a particular interest in pediatric, adolescent, and young adult populations. Snowball sampling, a purposeful non-probability sampling technique, was also used. The lead researcher [RU] initiated contact with all potential participants via email. If the participant expressed interest, the research associate [SM] reached out to the participant, provided an overview of the study nature and rationale, and arranged a time to complete the informed consent discussion and interview. A second follow-up email was sent if a potential participant failed to respond to the initial contact within one week.

Survivors and family/friend caregivers were recruited by distributing study information to cancer patient, survivor, and caregiver organizations, including support groups and networks, and patient advocacy organizations. The team intentionally contacted organizations and groups that served diverse populations (i.e., different cancer types and ages) or geographies (e.g., support groups in rural areas). Interested persons were asked to contact the research associate via phone or email for more information. Subsequently, the research associate provided an overview of the study and its rationale, and arranged a time to complete the informed consent discussion and interview.

### 2.2. Data Collection

The data for this paper are part of a larger dataset collected for a prior study [17]. Data were collected via one-on-one, semi-structured telephone interviews between the research associate and participant. The interviewer [SM] had no prior relationship with any participants. Interviews were audio-recorded and transcribed verbatim by an experienced transcriptionist. All interviews were conducted in 2020, and lasted between 15–54 min. There were no repeat interviews. The Interview Guide in Supplementary Materials was guided by the study objectives and developed based on guidance from Patton [18] and Rubin and Rubin [19]. This guidance included how to devise questions and order them,

how to structure questions to show respectful behavior toward participants, and how to forge conversational interviews. It was further informed by the need domains that have been assessed by the Pan-Canadian Transitions in Care Study to question participants on the range of potential needs experienced by cancer survivors as they transition from active treatment to long-term follow-up care [2]. Participants were asked to reflect on what they believe to be most important after cancer treatment and describe what an ideal intervention during this time would look like. In effect, they were asked to "blue sky" an intervention to meet the needs of survivors after completing cancer treatment. Despite evidence that existing interventions address psychosocial needs (e.g., physical activity), the research team inquired broadly about "blue sky" interventions in an effort to uncover, where possible, novel insights into accommodations that are desired by survivors, or that they would benefit from after completing treatment. Recruitment continued until data saturation was achieved.

### 2.3. Data Analysis

To analyse the data, the research team used methods commonly used in descriptive qualitative research, including, coding, grouping, detailing, and comparing the data [16]. Prior to developing a codebook, the lead researcher [RU] and research associate [SM] independently reviewed and coded three transcripts to identify the primary concepts. Subsequently, a codebook was developed compiling deductive and inductive concepts. The research associate used this codebook to code additional transcripts, during which codes were expanded or merged as appropriate. After a review of codes by the research team, the research associate continued to code the remaining transcripts, meeting with the lead researcher regularly to review and question the coding. Themes were sorted and compared across stakeholder groups to identify commonalities across groups in terms of the interventions deemed important to meet survivors' and families' psychosocial needs after completing cancer treatment. Important intervention areas commonly discussed by participants were organized into five categories identified posteriori. The analysis was performed manually, with the assistance of qualitative software (NVivo) for data management and to facilitate the comparison and synthesis of codes.

### 3. Results

Forty-four participants participated in this study. Most were healthcare providers or decision-/policy-makers (*n* = 26; 59.1%), with cancer survivors or family/friend caregivers representing (*n* = 18; 40.9%) of the sample. The participants resided in different areas across Canada; 31.8% were from Western Canada (*n* = 14), 38.6% were from Central Canada (*n* = 17), and 29.5% were from Atlantic Canada (*n* = 13). The cancer types represented by survivors and family/friend caregivers were breast, colorectal, genitourinary, ovarian, melanoma, and hematologic, and more than half of these participants were between the ages of 40–64 (55.6%). More details on participant demographics can be found in [17]. Stakeholder-relevant interventions related to survivors' psychosocial needs were classified into the following five intervention categories: information provision, peer support, navigation, knowledge translation interventions, and caregiver-specific supports. Table 1 describes specific examples and modes of delivery for each intervention category.

### 3.1. Informational Interventions

Interventions that deliver relevant, 'when needed' information were deemed critical in helping people plan for life after treatment, knowing what life would look like after treatment, and adjusting to one's new normal. As stated by one participant, "In my mind, knowledge is probably the most important thing when you are faced with cancer, and of course with follow-up thereafter" [P18 Survivor]. Informational interventions were deemed necessary because people desire to know what to expect in the next phase of their cancer care and require reassurance and ongoing support, particularly at a time when they find themselves less connected to their care team. As one participant described:

"Just like figuring out, like I think it wasn't until I was actually six months finished treatment that I actually, like took a breath and was like, 'oh, that just happened to me.' And so I think the fact that then I was like, 'okay, well, I'm not going to see my doctor for another three months, I don't really know who to contact.' ... When I went to my family doctor, no one really had anything to offer that was very cancer-specific ... But I still just needed so much more support processing what had happened to me, and for it to be cancer-specific."

(P27 Survivor)

**Table 1.** Specific examples of interventions by category.

| Intervention Category | Forms or Modes of Delivery |
|---|---|
| Informational interventions | Mobile health applications, in-person workshops, survivorship care plans, virtual information platforms |
| Peer support interventions | In-person or virtual networks and platforms to connect with peers and share experiences |
| Navigation interventions | Professional and peer-led navigation programs, and nurses with navigation roles and competencies |
| Knowledge translation interventions | Automatic referral systems to counselling, lifestyle, and rehabilitation interventions, and active strategies to implement practice guideline recommendations |
| Caregiver-specific interventions | Parallel caregiver resources and services to provide emotional support, program or clinic for caregivers and families, and caregiver-specific informational interventions |

Many participants also discussed how informational interventions are necessary to help survivors adjust to their new post-treatment identities and social roles. Finally, healthcare providers discussed the way that early and ongoing access to information about one's cancer and its treatment can reduce anxiety at the end of treatment and empower survivors to manage their ongoing health and wellbeing. This was exemplified by one participant who said:

"I think I would love to see survivorship start from the time of their diagnosis. So, our aim at our program, we have this new computer system ... So they can actually view their care plan. They can't do it right now. We'd love for them to be able to view their care plan and see, okay, they've had this chemo, these are some of the symptoms they felt, they'd been dealt with, this is how you can improve those symptoms. ... They can say, 'Okay, survivorship is coming. What is available to me? So these are some of my symptoms. These are where my resources I can go to ... " So all along, they can feel more a part of the program. And, at the end, understanding that they are going to be transitioned to the primary care provider at the end of their treatment, and not being anxious. Because that's a lot of the education piece—that they also feel part of it."

(P17 Healthcare provider (oncology nurse))

Suggested interventions include mobile applications, workshops, survivorship care plans, and online platforms that allow survivors to retrieve just-in-time resources. Participants also discussed the importance of personalizing how information is delivered, citing potential differences in modality preferences based on individual needs and circumstances (e.g., age and comfort with technology).

*3.2. Peer Support Interventions*

Most participants discussed the lack of support provided to survivors and caregivers during the transition from active treatment to routine follow-up care. As participants emphasized, "There's no, like, support transitioning back into my regular life" [P27 Survivor] and "It's just kind of like, 'Good luck. Let us know if anything else happens'" [P29 Caregiver]. As a result, communication with peers about what to expect after treatment, and

ongoing peer contact and support, were viewed as particularly valuable to help survivors adjust to a new normal after treatment, and to cope with ongoing and emerging health concerns. As one participant expressed:

> "I think that peer support is a good one. . . . Now after I've recovered, I think that that's more what I'm seeking, is sort of to talk to people who have had a similar experience or who have tried things and that have worked and haven't worked. Because it is such a difficult experience to share with people who don't understand."
>
> (P27 Survivor)

Caregivers also emphasized how peer support benefits psychosocial health recovery and is different from the support of one's family/friends:

> "Sharing what other people's stories are. Because my mom really only had hers. . . . I think having some kind of like support group for returning to work and the struggles that you go through. Going back to work is really challenging for her. And I mean luckily, we have a very supportive family. But that's very different than having a group of people that have been through similar things and can relate."
>
> (P29, Caregiver)

While peer support was more often discussed by survivors and caregivers, the importance of peer contact and support was also described by numerous decision-/policy-makers. As stated by one participant:

> "I think having some network or system electronically where they could communicate with each other and share their stories and provide support for each other, especially in the young adult group . . . So I think a lot of the patients feel that their experience makes them somehow different than their peers. And having more peer connectedness, I think would help them."
>
> (P8, Decision-/policy-maker)

### 3.3. Navigation Interventions

The need for navigation was discussed, of both a professional and peer-led character, to help survivors navigate the follow-up period and facilitate access to the resources and support they require, so as to manage long-term effects and optimize health recovery after treatment. This was deemed particularly important given that many of these resources and supports fall outside of the walls of cancer clinics/centres, as another participant provided:

> "And the navigators, I did use the navigator in the [Community X] hospital years after. And I can't remember what it was for. But it was so good to know that here was somebody locally who knew the local community and could pinpoint where the help could come from. That would be so good."
>
> (P12, Survivor)

Navigation was also seen as an important emotional and psychosocial support that can help survivors adjust to life beyond cancer, which was noted by P2:

> "If I could blue sky it, we would have a transition plan including navigation. It could be virtual navigation, it could be a function provided by a person. It could be peer navigation. Because we have tremendous success with all of them. And if we could blue sky it, we would have all of them for survivorship. Because people certainly choose what works best for them. . . . And the greatest benefit I see is how it improves people's preparedness to cope with what they're facing."
>
> (P2, Decision-/policy-maker)

Many participants felt that survivors who are able to access navigation services have better outcomes than those who do not. One participant described this as:

> "I deal with navigational patients all over Canada . . . And I find the ones that have like a nurse to help navigate them through the system do far better than those that do not. So having a navigation system and points of contact where they can speak to somebody is very important."
>
> (P17, Healthcare provider (oncology nurse))

### 3.4. Knowledge Translation Interventions

Among the participants, many recognised that evidence-based interventions are not integrated into survivorship care in a timely or consistent way. Consequently, all stakeholder groups discussed the need for knowledge translation interventions to ensure that interventions that are already known to be effective are well integrated into post-treatment follow-up care so that survivors benefit from these evidence-based programs and support. Many healthcare providers and decision-makers acknowledged that practice guidelines exist for survivorship care (e.g., guidelines related to physical activity and psychosocial oncology), yet they believe such guidelines to be inconsistently implemented in our health care system. Survivors and caregivers expressed this sentiment in terms of "gaps" in services and supports that are known to benefit one's recovery, specifically as they discussed their lack of access to rehabilitation and physical activity programs, professional psychology services, and self-management programs. As one participant commented when speaking about professional mental health services, "so, there's a huge gap, a huge gap there for depression and a lot of other issues" (P4, Survivor). Healthcare providers and decision-makers also discussed gaps in evidence-based services and supports for their survivor populations. One participant described the lack of evidence-based programs in the following way:

> "I think there's a piece there too, where with exercise, we should be saying something very similar. And with nutrition, something very similar. You know, all the way through you're promoting healthy lifestyle in people long before they're finished active treatment and moving forward. Because I think that's a huge gap that right now, we don't do very much of it. They finish active treatment. They have their ongoing follow-up with the physician based on whatever the disease is that they've been dealing with . . . But they don't have that easy access necessarily to counselling from a dietitian or counselling from a physiotherapist or somebody that's there to help support and coach and counsel them related to transitioning back into the workplace. You know, now that you've got these deficits, it is going to be harder for you to do the job you did before you were diagnosed."
>
> ((P41 Healthcare provider (nurse practitioner))

### 3.5. Caregiver-Specific Interventions

Caregiver participants discussed how their experiences were different to those of the patient/survivor and how they often feel overwhelmed during and after treatment. One caregiver spoke of her emotional state in the following way:

> "I think, you know, I would say I experienced some trauma. Like I said, I hated that cancer centre. I went up, I was supporting [partner]. But it's like I wanted someone to say to me, '[participant], I know that this is absolutely brutal.' . . . It didn't seem real to me. It just seemed, it seemed like a mechanical assembly line, quite frankly. . . . I mean I would see even nurses taking care of [partner], and they had a nice rapport with him, and they were, like they were very efficient and very caring. And I, like it's horrible to say, hated them."
>
> (P42 Caregiver)

Consequently, all caregiver participants discussed the need for parallel caregiver resources and services to provide emotional support in the post-treatment follow-up period. As described by one participant:

"There was a lot of like emotional things for me that I never really got any support for on how to navigate those emotions and how to kind of process that like your mom's had cancer and had to go through the treatment. And kind of like, I think, that trauma that comes from a cancer diagnosis. So for me, it was a lot of that. I think there needs to be some sort of like, either like a program or a clinic or something that offers emotional support to cancer survivors and their families and their caregivers to navigate all of that emotional side of it once someone's gone through cancer. Because like cancer doesn't end when the treatment ends. Like there's a lot of the post-cancer stuff. It changes your life forever."

(P32, Caregiver)

Most frequently, participants cited wanting caregiver-specific informational and peer support interventions, for example:

"So, there was really a lack of information for me as a caregiver as kind of like what are the next steps as far as like, you know, your cancer treatments are done. . . . I wish there was some kind of like follow-up appointment for the family as well where it could be like clearly indicated like this is what to expect over the next year or two years, this is what you can be watching for."

(P29, Caregiver)

"You know, whether there was someone you could talk to, you know, as a caregiver . . . Or online supports . . . I think it's good to see that connectivity across distance for people to be able to connect. . . . To apply it into a caregiver scenario would, I think, be very, very helpful."

(P44 Caregiver)

## 4. Discussion

This study sought to capture stakeholders' views on interventions that are most desired and best suited to meet survivors' and their families' psychosocial concerns after completing cancer treatment. To do so, stakeholders from across Canada were asked to describe interventions they felt were most likely to address survivors' ongoing concerns during the follow-up care period. The important interventions were classified into the following five categories: information provision, peer support, navigation, knowledge translation, and caregiver-specific supports/resources. This work is intended to inform future survivorship intervention research. As a starting point, researchers should use these findings to develop and test strategies to deliver more personalized and timely information to survivors, rigorously test the effectiveness and cost-effectiveness of various navigation models during the post-treatment survivorship period, and facilitate the translation of existing research into sustainable programming and practice, including caregiver-specific interventions as these individuals often undergo immense emotional distress during and after their loved one's cancer diagnosis and treatment.

Interventions to address the informational and psychosocial needs of survivors have been deemed as urgent research priorities in Canada [15]. Indeed, the lack of relevant and timely information has been cited as a long-standing gap in cancer survivors' care in Canada [10,13,14]. Participants in this study suggested interventions such as mobile applications, in-person workshops, and survivorship care plans as mechanisms to provide survivors with the information they require to manage their psychosocial wellbeing. Some of these interventions have been extensively studied. For example, randomized controlled trials (RCTs) on the effectiveness of survivorship care plans are reasonably well-documented yet have yielded mostly unsubstantial or mixed results [20]. Conversely, qualitative and survey data suggest survivors do benefit from this information [21,22]. For example, a population-based survey study in Nova Scotia, Canada, showed that of the allocation of survivorship care plans was significantly associated with receiving timely support to meet one's informational and psychosocial needs in the 1–3 years after cancer treatment [23]. It is noteworthy that the primary outcome of RCTs tends to be quality of

life, a complex construct that is likely influenced by more than the provision of information. This points to the need for researchers to thoughtfully consider modes of intervention before selecting the outcomes it might reasonably impact.

Research has shown that cancer-patient navigation improves the timeliness of care and the uptake of evidence-based procedures and reduces healthcare costs [24–30]. In fact, the coordination of care via professional navigation (e.g., nurse or social work) is an accreditation standard for breast centres in the United States [31]. However, much of the evidence comes from cancer screening, diagnosis, and treatment, with only limited evidence regarding the effectiveness of navigation during the survivorship period. A population-based study in Nova Scotia, Canada, found that accessing a cancer-patient navigator after completing treatment was associated with having one's psychosocial needs met [32]. Similarly, peer support has been cited as both a valued and desired tool to support coping and recovery during the cancer journey [33–35]. An RCT of peer support for men who had received radical prostatectomy for prostate cancer enhanced self-efficacy and reduced depression compared to men in the control group [36].

Participants in this study described the need for knowledge-translation interventions (also known as implementation strategies) to integrate effective interventions into routine practice. Implementation science, a field of science that employs rigorous methods to design and test strategies to more effectively integrate, spread, and sustain research evidence into clinical and public health practice and policy [37], must become more commonplace in cancer survivorship research. This has been previously recognized as a priority by cancer research funders and others in Canada [15], with a recent national collaborative funding competition requiring both intervention research and implementation science to advance cancer survivorship care and outcomes [38]. Achieving developments in this area, however, will require that clinical, health service, and population health researchers to develop their understanding of implementation science, including its theoretical underpinnings and approaches to applying theory, study designs (e.g., hybrid effectiveness-implementation trials [39], mixed methods), and transdisciplinary modes of working, wherein research teams include not only researchers but also include decision-/policy-makers, clinicians, public health professionals, and patients, clients, families, and citizens. There is also a need to focus on sustainability because even when evidence-based practices are implemented, they often fail to become integrated into the long-term routines of organizations [40–42]. A recent study of 25 evidence-based cancer survivorship interventions, implemented in Canadian jurisdictions, identified a range of factors influencing their sustainability [43]. This work should be leveraged and built on to understand how best to sustain interventions after initial implementation so that survivors receive the care and support they need to optimize health recovery.

Relatedly, there is a pressing need to ensure that interventions that are known to improve caregivers' experiences and outcomes become a routine part of person-centred cancer care. In this study, all caregivers discussed the need for caregiver-specific interventions, whereas participants in other stakeholder groups did not identify this as an important area. However, we are aware that caregivers often experience a considerable physical and emotional toll while caring for a loved one with cancer. For example, they report numerous physical effects as a result of the stress of caregiving, including fatigue, insomnia, and overall diminished health, as well as psychosocial needs, including anxiety, distress, uncertainty, social isolation, and loss of identity [44–50]. These effects begin at the time of diagnosis and often persist to end of life or during long-term survival. Systematic reviews examining caregiver interventions have demonstrated that caregiver-specific interventions can result in the improvement of coping skills, increased self-efficacy, and an improved quality of life [51,52]. While generally beneficial, a recent review on caregiver-specific interventions in oncology found that few studies reported on their potential of being implemented more broadly [53]. Specifically, the authors found that few studies were designed in a way that would maximize their potential for implementation outside of a trial setting, and that only a few studies reported information relevant to implementation. Both are critical if we are

to adopt and successfully implement efficacious interventions in real world scenarios to benefit caregivers of cancer patients and survivors [54].

This study has several limitations. First, this study was undertaken in Canada, and therefore does not account for the desires and requirements, from a psychosocial perspective, in jurisdictions for which the health care system differs substantially from Canada. Second, asking participants about ideal interventions to improve survivors' outcomes may be an abstract request, leading participants to describe interventions they are already aware of as opposed to encouraging creativity in their thinking. Finally, participants often discussed interventions in terms of their form and delivery modes, and less often in terms of the specific content such forms should include. Nonetheless, we interviewed a broad range of stakeholders from across Canada who had varied experiences with cancer survivorship (e.g., cancer types, subpopulations). Therefore, the findings provide guidance for researchers and cancer programs in terms of future directions in cancer survivorship research and programming.

## 5. Conclusions

This study identified five categories of interventions that stakeholders view as important to address important psychosocial needs in the post-treatment survivorship period. The five categories are as follows information provision, peer support, navigation, knowledge translation interventions, and caregiver-specific supports. These findings have implications for both future research and practices, namely, to develop interventions that deliver timely and relevant information about the post-treatment period, and provide knowledge-translation interventions that integrate effective tools and programs into survivorship care. Both factors are critical toward improving survivors' experiences and psychosocial outcomes as they transition from primary treatment to follow-up care.

**Supplementary Materials:** The following are available online at https://www.mdpi.com/article/10.3390/curroncol28060416/s1, Document S1: Interview Guide.

**Author Contributions:** Conceptualization: R.U., C.K., J.S., G.A.P., D.H. and E.G.; methodology: R.U., S.M. and C.K.; formal analysis: R.U. and S.M.; writing—original draft preparation: R.U. and S.M.; writing—review & editing: C.K., J.S., G.A.P., D.H. and E.G.; Funding Acquisition: R.U. All authors have read and agreed to the published version of the manuscript.

**Funding:** This research was funded by a Foundation Scheme Grant from the Canadian Institutes of Health Research, grant number FDN-154323.

**Institutional Review Board Statement:** The study was conducted according to the guidelines of the Declaration of Helsinki, and approved by the Research Ethics Board of Nova Scotia Health (protocol number 1024142, approved 18 December 2018).

**Informed Consent Statement:** Informed consent was obtained from all participants involved in the study.

**Data Availability Statement:** The data presented in this study are available on request from the corresponding author. The data are not publicly available due to privacy and confidentiality considerations.

**Acknowledgments:** We gratefully acknowledge all of the study participants, who took the time to participate in this study. We are also grateful for the support of Margaret Jorgensen who assisted with study operations.

**Conflicts of Interest:** The authors declare no conflict of interest. The funder had no role in the design, execution, interpretation, or writing of the study.

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
