# Peer review of "Stakeholder-Identified Interventions to Address Cancer Survivors’ Psychosocial Needs after Completing Treatment"

_curroncol, doi:10.3390/curroncol28060416_

Round 1
Reviewer 1 Report
Thank you for letting med review this interesting and well-written manuscript. The manuscript adress an important and understudied topic- psychosocial support after cancer treatment.
Please highlight citations throughout the manuscript from the start of the citation- not only at the end.
A limitation of this manuscript is that the topics and citations are described at a high level, with less specific information about the needs for psychosocial interventions after cancer treatment. Th current focus is more on the format of the psychosocial support (group/app/etc) - and less on the content.
regarding limitations p 7: limited representativeness may not be a relevant limitation in a qualitative study.
Author Response
Comment: Thank you for letting me review this interesting and well-written manuscript. The manuscript address an important and understudied topic- psychosocial support after cancer treatment.
Response: Thank you for this comment.
Comment: Please highlight citations throughout the manuscript from the start of the citation- not only at the end.
Response: Thank you for this comment. We have formatted our citations according to Current Oncology’s style requirements. However, if the Editor would like the citations placed elsewhere, please let us know.
Comment: A limitation of this manuscript is that the topics and citations are described at a high level, with less specific information about the needs for psychosocial interventions after cancer treatment. The current focus is more on the format of the psychosocial support (group/app/etc) - and less on the content.
Response: We agree this is one limitation of our dataset and resulting manuscript. Therefore, we have included the following in the limitations paragraph of the Discussion section (pg 8 of revised manuscript):
Finally, participants often discussed interventions in terms of their form and delivery modes, and less in terms of the specific content those forms should include.
Comment: regarding limitations p 7: limited representativeness may not be a relevant limitation in a qualitative study.
Response: We agree with the Reviewer. We have reworded this statement to now say (pg 8 of revised manuscript):
First, this study was undertaken in Canada, and therefore what is needed or desired from a psychosocial perspective may be different in jurisdictions whose health care systems differ substantially from Canada.
Reviewer 2 Report
This paper covers an important topic by assessing stakeholder perceptions of interventions needed for cancer survivorship. The study could be strengthened by developing a stronger framing in the introduction (a more comprehensive review of survivorship needs), more details in the methods (e.g., interview length, ethics review), and better organization in the results section (e.g., subheadings, comparing results across stakeholder groups). Specific suggestions are provided below:
- In the introduction, it would be great to expand on some of the specific issues cancer survivors experience. For example, the intro mentions 'physical needs' and 'economic needs'. The second paragraph elaborates on what is meant by psychosocial needs but does not provide more detail on the other needs mentioned in the first paragraph.
- In the methods, it says that purposive sampling was used to get a variety of participants based on 'cancer type', 'residence', etc. Can you go into more detail to say how your recruitment strategy ensured an adequate mix of these factors? For example, did you have to target rural care systems to ensure you recruited some rural patients?
- It says the interview guide was based on guidance from Patton and Rubin and Rubin. Can you describe in the text what these citations refer to?
- Can you describe how you made sure the interview guide covered all of the potential needs a survivor might have (e.g., was there a framework used)?
- Can you add what type of ethics review (e.g., IRB) was done to approve the study in the methods section?
- Can you add how long the interviews were to the methods section?
- Can you add what year the interviews were conducted to the methods?
- It may be helpful to add subsections to the methods section (e.g., study sample, recruitment, data collection, analyses)
- It may be helpful to ensure that your study conforms to a checklist for qualitative research such as the COREQ
- For the results, in the 'informational interventions' is there a way to break up this text with subheadings? Some of the information is about content vs delivery and it gets confusing switching between the two.
- Under the knowledge translation section, it isn't clear if some of the text is about knowledge translation. For example, under knowledge translation interventions, you discuss need for healthy lifestyle counseling (exercise and nutrition). It doesn't become clear until you get to the end of the section where you say guidelines exist but are not implemented. So maybe relabel the section or make the knowledge translation gap more clear in the beginning of the section.
- I think the study stratification method (e.g, trying to get mix of rural/urban) doesn't carry through in the results. Did you observe differences based on cancer type, rural/urban, or adult/pediatrics (the factors you were trying to ensure had sufficient sample diversity)?
- Similarly, did you observe any differences by stakeholder type (e.g., caregiver vs. healthcare provider)?
Author Response
Comment: This paper covers an important topic by assessing stakeholder perceptions of interventions needed for cancer survivorship. The study could be strengthened by developing a stronger framing in the introduction (a more comprehensive review of survivorship needs), more details in the methods (e.g., interview length, ethics review), and better organization in the results section (e.g., subheadings, comparing results across stakeholder groups).
Response: Thank you for this comment and for your suggestions. See below for our responses.
Comment: In the introduction, it would be great to expand on some of the specific issues cancer survivors experience. For example, the intro mentions 'physical needs' and 'economic needs'. The second paragraph elaborates on what is meant by psychosocial needs but does not provide more detail on the other needs mentioned in the first paragraph.
Response: We have now added the following statement to the introduction to describe prevalent ongoing needs (pg 1 of revised manuscript):
The most prevalent of these needs were: fatigue, changes in sexual function, and changes in memory and concentration (physical); worry about cancer recurrence, changes in sexual intimacy, and depression (emotional); and challenges returning to work/school, challenges getting to/from appointments, and paying for health care (practical).
Comment: In the methods, it says that purposive sampling was used to get a variety of participants based on 'cancer type', 'residence', etc. Can you go into more detail to say how your recruitment strategy ensured an adequate mix of these factors? For example, did you have to target rural care systems to ensure you recruited some rural patients?
Response: Thank you for this comment and the opportunity to clarify our sampling frame. For healthcare providers and decision-makers, we purposively sampled based on cancer type, place of work or residence (urban/rural/remote), and special populations (e.g., pediatric, adolescent, and young adults). That is, we made sure we identified and contacted healthcare providers and decision-makers who specialized in different cancer types, who worked in urban and rural settings, and who worked with or had a particular interest in pediatric and AYA populations. For survivors/caregivers, we were unable to identify people and contact them directly, so this was certainly more challenging. However, we reached out to cancer patient, survivor, and caregiver organizations and groups that also varied by or served these same varied characteristics. So, we reached out to organizations and groups that served different populations (i.e., different cancer types and ages) or that served rural populations (e.g., support groups in rural areas). We have now included the following sentences to expand on our recruitment strategy (pg 2 of revised manuscript):
To achieve variation on the sampling characteristics, the research team identified and contacted individuals who specialized in different cancer types, who worked in urban and rural settings, and who worked with or had a particular interest in pediatric, adolescent, and young adult populations.
The team purposely contacted organizations and groups that served diverse populations (i.e., different cancer types and ages) or geographies (e.g., support groups in rural areas).
Comment: It says the interview guide was based on guidance from Patton and Rubin and Rubin. Can you describe in the text what these citations refer to?
Response: Yes, the interview guide (and indeed the interview itself) was based on practical guidance from both sources. For example, Rubin and Rubin discuss the importance of forging a conversational partnership between the researcher and the participant, and provide practical advice around how to do so. This guidance ranges from how to structure the questions to how to show respectful behavior toward participants to not worrying about missteps in questioning because there are few “deal-breaking” mistakes when it comes to interviewing. Patton provides constructive guidance on how to devise questions (singular questions are encouraged), how to order them, and how to ensure they resonate with participants. We have updated the text in the revised manuscript (pg 2) to describe this guidance:
This guidance included how to devise questions and order them, how to structure questions to show respectful behavior toward participants, and how to forge conversational interviews.
Comment: Can you describe how you made sure the interview guide covered all of the potential needs a survivor might have (e.g., was there a framework used)?
Response: Thank you for this important question. Our interview guide was quite simple. However, we used the domains assessed by the Pan-Canadian Transitions in Care Study to probe the range of potential needs (see: https://www.ncbi.nlm.nih.gov/pmc/articles/PMC6597588/). These domains were physical, emotional, and practical, and were based on the LiveStrong and Cancer Survivors Unmet Needs Measure (CaSUN) surveys, both of which have been validated. We have added the following sentence to the methods section (pg 2-3 of the revised manuscript):
It was further informed by the need domains assessed by the Pan-Canadian Transitions in Care Study to probe the range of potential needs experienced by cancer survivors as they transition from active treatment to long-term follow-up care.
Comment: Can you add what type of ethics review (e.g., IRB) was done to approve the study in the methods section?
Response: It was an institutional review by the Research Ethics Board at Nova Scotia Health. This was indicated at the end of the manuscript but if there are additional details requested, please let us know.
Comment: Can you add how long the interviews were to the methods section?
Response: Thank you. We have added this to the methods section (pg 2 of the revised manuscript), but we also completed the COREQ reporting checklist for qualitative studies, which we have submitted as a supplementary file. This file contains this information as well as additional details around methodological aspects.
Comment: Can you add what year the interviews were conducted to the methods?
Response: All interviews occurred in 2020. This has been added to the manuscript (pg. 2 of revised manuscript).
Comment: It may be helpful to add subsections to the methods section (e.g., study sample, recruitment, data collection, analyses).
Response: Thank you for this suggestion. We have now added subsections to the methods section.
Comment: It may be helpful to ensure that your study conforms to a checklist for qualitative research such as the COREQ
Response: Thank you for this comment. We have completed the COREQ checklist and submitted it as a supplementary file to our resubmission.
Comment: For the results, in the 'informational interventions' is there a way to break up this text with subheadings? Some of the information is about content vs delivery and it gets confusing switching between the two.
Response: You are correct, some of the information is about content whereas some is about delivery. We have now revised this theme so that content is discussed first and format/modality at the end.
Comment: Under the knowledge translation section, it isn't clear if some of the text is about knowledge translation. For example, under knowledge translation interventions, you discuss need for healthy lifestyle counseling (exercise and nutrition). It doesn't become clear until you get to the end of the section where you say guidelines exist but are not implemented. So maybe relabel the section or make the knowledge translation gap more clear in the beginning of the section.
Response: This theme refers to interventions (also known as implementation strategies) to facilitate the translation of research evidence (what we know works) into practice (what we do). We have moved around some statements in this section in an attempt to maximize clarity.
Comment: I think the study stratification method (e.g., trying to get mix of rural/urban) doesn't carry through in the results. Did you observe differences based on cancer type, rural/urban, or adult/pediatrics (the factors you were trying to ensure had sufficient sample diversity)?
Response: This is an excellent question. When we analyzed desired interventions (which is what we present in this manuscript), we did explore whether differences existed based on these factors. We found that this was not the case. All intervention types were discussed and prioritized across patient populations. However, in a prior manuscript using data from the same dataset that described prioritized outcomes, we did find differences in terms of important outcomes based particularly on age.
Comment: Similarly, did you observe any differences by stakeholder type (e.g., caregiver vs. healthcare provider)?
Response: The differences across stakeholder type that we found really related to caregivers. Not one participant from the non-caregiver groups discussed the need for caregiver-specific interventions. However, this was emphasized, in depth, by all caregiver participants. This is an important finding and we have added the following statement in the discussion section (pg. 7 of revised manuscript) to emphasize this:
In this study, all caregivers discussed the need for caregiver-specific interventions whereas participants in the other stakeholder groups did not discuss this as an important area of need.
Reviewer 3 Report
Overall comments
This is a meaningful study that attempted to identify post-treatment interventions addressing cancer survivors’ psychosocial needs through the perspectives of relevant stakeholders. Findings from this study will help guide the development and implementation of psychosocial interventions that address the five stated categories. The large sample size for this qualitative study is a major strength. I have some minor comments below for consideration.
Introduction
- The extent of unmet psychosocial needs, and need for intervention research, is well described in the introduction. Could the authors elaborate on some examples of psychosocial needs (reintegration after cancer treatment, fear of cancer recurrence, and anxiety were listed as most common among these participants in Ref 19).
- Similarly, could the authors provide some examples of how interventions may be able to address several needs at once (providing justification for why this study looks broadly at blue-sky interventions for psychosocial outcomes not at interventions with specific outcomes in mind such as fear of recurrence or anxiety).
- There are various existing psychological and physical activity interventions that have been developed and tested to address cancer survivors’ psychosocial needs post-treatment, so could the authors provide some justification as to what has been lacking from previous research and what gap this study is addressing. I assume it is the focus on stakeholders needs and insights into translation strategies but this could be made clearer.
Methods
- The methods described are appropriate. Could the authors clarify whether purposive sampling was used just for the cancer survivors or other professional groups too (providers could also be recruited from rural/urban areas and from different specialties with regards to pediatric/adult, cancer types).
- It would be helpful to state that this data comes from a previously published study with project methods described elsewhere (19) and explain the different aims and data used in this manuscript.
Results
- Grammatical edit – “family/friend caregivers representing 41% of the sample (n=18)” – Line 104-105.
- Be consistent with decimal place in percentages.
- For international readers, could the authors explain whether Western, Central and Atlantic Canada represents the urban/rural/remote locations. Or if this was not collected by participants perhaps remove from methods.
- It may be helpful to specify the type of healthcare provider in quotations if feasible, as oncologists may have quite different views to those in primary care who have less experience with cancer.
- While the categories of interventions are well described and supported with participant quotations, it may be helpful to future researchers or policymakers to have an overview of the specific interventions/delivery modes that could be developed based on participant comments. Perhaps in a table such as the example below:
|
Intervention need |
Mode of delivery |
|
Informational interventions |
mobile applications, in-person workshops, and survivorship care plans |
|
Peer support interventions |
network or system electronically where they could communicate avenue for sharing stories |
|
Navigation interventions |
professional and peer-led navigation a transition plan including navigation a nurse to help navigate them through the system |
|
Knowledge translation |
easy access to counselling from a dietitian or physiotherapist implementation of evidence-based guidelines |
|
Caregiver |
parallel caregiver resources for emotional support a program or a clinic for caregivers and families caregiver-specific informational and peer 254 support interventions |
Discussion
- The authors did a good job of selecting a diverse range of participants and stakeholders. Is there any comment that could be discussed regarding differences in need or intervention methods among cancer types, subpopulations etc despite the smaller numbers.
- A number of references relating to caregiver anxiety, distress etc are more than 10 years old (47-49), are there some updated references that could be found?
References
- Reference 54 doesn’t have a year listed
Author Response
Comment: This is a meaningful study that attempted to identify post-treatment interventions addressing cancer survivors’ psychosocial needs through the perspectives of relevant stakeholders. Findings from this study will help guide the development and implementation of psychosocial interventions that address the five stated categories. The large sample size for this qualitative study is a major strength. I have some minor comments below for consideration.
Response: Thank you for this comment.
Introduction
Comment: The extent of unmet psychosocial needs, and need for intervention research, is well described in the introduction. Could the authors elaborate on some examples of psychosocial needs (reintegration after cancer treatment, fear of cancer recurrence, and anxiety were listed as most common among these participants in Ref 19).
Response: In response to a comment from Reviewer 2, we have now added the following statement to the introduction section to highlight prevalent ongoing needs in a national (Canadian) study on survivors’ needs after treatment. We hope this is sufficient in terms of providing examples of ongoing needs.
The most prevalent of these needs were: fatigue, changes in sexual function, and changes in memory and concentration (physical); worry about cancer recurrence, changes in sexual intimacy, and depression (emotional); and challenges returning to work/school, challenges getting to/from appointments, and paying for health care (practical).
Comment: Similarly, could the authors provide some examples of how interventions may be able to address several needs at once (providing justification for why this study looks broadly at blue-sky interventions for psychosocial outcomes not at interventions with specific outcomes in mind such as fear of recurrence or anxiety).
Response: This is a fair comment. Indeed, navigation and peer support have both been shown to address multiple needs of cancer survivors. Similarly, physical activity interventions address multiple needs of cancer survivors (e.g., fatigue, anxiety, and depression). We have now added the following statement to the methods section to provide further justification for our study (pg. 3 of revised manuscript):
Despite evidence that existing interventions address psychosocial needs (e.g., physical activity), the research team inquired broadly about “blue sky” interventions in an effort to uncover, where possible, novel insight into things that survivors desire or would benefit from after completing treatment.
Comment: There are various existing psychological and physical activity interventions that have been developed and tested to address cancer survivors’ psychosocial needs post-treatment, so could the authors provide some justification as to what has been lacking from previous research and what gap this study is addressing. I assume it is the focus on stakeholders needs and insights into translation strategies but this could be made clearer.
Response: Yes, there are certainly interventions already developed and tested to address psychosocial needs after treatment. However, as per our response to the above comment, we have now added a statement to the methods section (pg. 3 of revised manuscript):
Despite evidence that existing interventions address psychosocial needs (e.g., physical activity), the research team inquired broadly about “blue sky” interventions in an effort to uncover, where possible, novel insight into things that survivors desire or would benefit from after completing treatment.
Methods
Comment: The methods described are appropriate. Could the authors clarify whether purposive sampling was used just for the cancer survivors or other professional groups too (providers could also be recruited from rural/urban areas and from different specialties with regards to pediatric/adult, cancer types).
Response: Purposive sampling was used for both groups. See our response to a comment from Reviewer 2 above. We have now included the following sentences to expand on our sampling strategy (pg 2 of revised manuscript):
To achieve variation on the sampling characteristics, the research team identified and contacted individuals who specialized in different cancer types, who worked in urban and rural settings, and who worked with or had a particular interest in pediatric, adolescent, and young adult populations.
The team purposely contacted organizations and groups that served diverse populations (i.e., different cancer types and ages) or geographies (e.g., support groups in rural areas).
Comment: It would be helpful to state that this data comes from a previously published study with project methods described elsewhere (19) and explain the different aims and data used in this manuscript.
Response: Thank you for this suggestion. We have added the following statement to be clear that these data are part of a larger dataset for a prior study (pg 2 of revised manuscript):
The data for this paper are part of a larger dataset collected for a prior study.
Results
Comment: Grammatical edit – “family/friend caregivers representing 41% of the sample (n=18)” – Line 104-105.
Response: Thank you. Fixed.
Comment: Be consistent with decimal place in percentages.
Response: Done. Thank you.
Comment: For international readers, could the authors explain whether Western, Central and Atlantic Canada represents the urban/rural/remote locations. Or if this was not collected by participants perhaps remove from methods.
Response: This is an important suggestion. We did recruit from across Canada and purposively sampled from rural and urban regions. However, we have not reported on rural vs urban demographics simply because this becomes somewhat complicated in terms of clear delineation (e.g., people have moved locations, people may live in a northern urban setting but care for many rural/remote populations, etc). Therefore, we have left the results as is for now. However, if necessary, we can attempt to categorize this information.
Comment: It may be helpful to specify the type of healthcare provider in quotations if feasible, as oncologists may have quite different views to those in primary care who have less experience with cancer.
Response: We have now identified the healthcare providers in the three relevant quotations.
Comment: While the categories of interventions are well described and supported with participant quotations, it may be helpful to future researchers or policymakers to have an overview of the specific interventions/delivery modes that could be developed based on participant comments. Perhaps in a table such as the example below:
|
Intervention need |
Mode of delivery |
|
Informational interventions |
mobile applications, in-person workshops, and survivorship care plans |
|
Peer support interventions |
network or system electronically where they could communicate avenue for sharing stories |
|
Navigation interventions |
professional and peer-led navigation a transition plan including navigation a nurse to help navigate them through the system |
|
Knowledge translation |
easy access to counselling from a dietitian or physiotherapist implementation of evidence-based guidelines |
|
Caregiver |
parallel caregiver resources for emotional support a program or a clinic for caregivers and families caregiver-specific informational and peer 254 support interventions |
Response: Thank you for this suggestion. We have now added a table to provide examples for each intervention type.
Discussion
Comment: The authors did a good job of selecting a diverse range of participants and stakeholders. Is there any comment that could be discussed regarding differences in need or intervention methods among cancer types, subpopulations etc despite the smaller numbers.
Response: This is an excellent question. As we stated above in response to a comment from Reviewer 2, when we analyzed desired interventions (which is what we present in this manuscript), we did explore whether differences existed based on these factors. We found that this was not the case. All intervention types were discussed and prioritized across patient populations. However, in a prior manuscript using data from the same dataset that described prioritized outcomes, we did find differences in terms of important outcomes based particularly on age.
Comment: A number of references relating to caregiver anxiety, distress etc are more than 10 years old (47-49), are there some updated references that could be found?
Response: The Reviewer is correct, many of these references are older. However, two of the reviews provided (#s 53 and 54) are from 2017 and 2019. We hope these are sufficiently recent.
References
Comment: Reference 54 doesn’t have a year listed
Response: Fixed. Thank you.
Round 2
Reviewer 2 Report
The authors have sufficiently address my comments